# Sociodemographic predictors of the association between self-reported sleep duration and depression

**Mitha Al Balushi**[1,2]*, **Amar Ahmad**[1], **Sara Al Balushi**[1,3], **Sayed Javaid**[4], **Fatma Al-Maskari**[2], **Abdishakur Abdulle**[1], **Raghib Ali**[1,5]*

**1** Public Health Research Center, New York University-Abu Dhabi, Abu Dhabi, United Arab Emirates, **2** Institute of Public Health, College of Medicine and Health Sciences, United Arab Emirates University, Alain, United Arab Emirates, **3** Lancaster University, Lancaster, United Kingdom, **4** Department of Psychiatry and Behavioral Sciences, College of Medicine and Health Sciences, United Arab Emirates University, Alain, United Arab Emirates, **5** MRC Epidemiology Unit, University of Cambridge, Cambridge, United Kingdom

* ma4643@nyu.edu (MAB); ra107@nyu.edu (RA)

**Data Availability Statement:** Data are from the United Arab Emirates Healthy Futures (UAEHFS) study. A de-identified data set can be shared

## Abstract

A growing interest has been recently reported in exploring sleep duration within psychology context in particular to its relation to some mental chronic diseases such as depression. The aim of this study is to investigate the association between self-reported sleep hours as an outcome and self-perceived depression among Emirati adults, after adjusting for sociodemographic factors such as age, gender, marital status, and employment status. We performed a cross-sectional analysis using 11,455 participants baseline data of the UAE Healthy Future Study (UAEHFS). Univariate and multivariate logistic regression models were performed with self-reported sleep hours as an outcome. The predictors were the self-reported depression by measuring the PHQ-8 score, sociodemographic factors (age, gender, marital status, and employment status) Odds ratios with 95% confidence intervals (CI) were reported. In a sensitivity analysis, a multivariate imputation by chained equations (MICE) procedure was applied with classification and Regression Trees (CART) to impute missing values. Overall, 11,455 participants were included in the final analysis of this study. Participants' median age was 32.0 years (Interquartile-Range: 24.0, 39.0). There were 6,217 (54.3%) males included in this study. In total, 4,488 (63.6%) of the participants reported sleep duration of more than 7 hours. Statistically significant negative association was observed between the total PHQ-8 score as a measure for depression and binarized self-reported sleep, OR = 0.961 (95% CI: 0.948, 0.974). For one unit increase in age and BMI, the odds ratio of reporting shorter sleep was 0.979 (95% CI: 0.969, 0.990) and 0.987 (95% CI: 0.977, 0.998), respectively. The study findings indicate a correlation between self-reported depression and an increased probability of individuals reporting shorter self-perceived sleep durations especially when considering the sociodemographic factors as predictors. There was a variation in the effect of depression on sleep duration among different study groups. In particular, the association between reported sleep duration and reported depression, students and unemployed individuals have reported longer sleep hours as compared to employed participants. Also, married individuals reported a higher percentage of

subject to the policies of the approving ethics committee and the data access policy of the UAEHFS. The New York University Abu Dhabi IRB approved informed consent form described how participant data would be shared with other researchers. The consent form states that researchers who are interested in accessing study data will contact the data access/ethics committee to be granted access to the data. Once approved, de-identified data can be made available. Researchers who meet the criteria for access to confidential data may contact the IRB at IRBnyuad@nyu.edu to gain access to the data.

**Funding:** This article is based upon work supported by Tamkeen under Research Institute, New York University in Abu Dhabi Grant No: G1206. This research was approved by the New York University in Abu Dhabi Research Ethics Committee (NYUAD/REC), (REF: 0072017R to RA) and the Institutional Review Board of Abu Dhabi Department of Health (DOH), (REF: DOH/HQD/2020/516 to RA). All methods were performed in accordance with the relevant guidelines and regulations. Written informed consent was obtained from all study participants prior to the start of data collection. Please address all correspondence concerning this manuscript to me at ma4643@nyu.edu, ra107@nyu.edu.

**Competing interests:** The authors have declared that no competing interests exist.

longer sleep duration as compared to single and unmarried ones when examined reported depression as a predictor to sleep duration. However, there was no gender differences in self-perceived sleep duration when associated with reported depression.

## Introduction

Sleep deficiency is evidenced to lead to lower well-being, physical and mental health disorders, disability, and loss of productivity [1, 2]. Insufficient sleep duration is connected to cardiovascular diseases such as heart disease, elevated blood pressure, diabetes mellitus, obesity [3]. According to the Centers for Disease Control and Prevention, about 1 in 3 adults in the United States reported not getting enough rest or sleep every day [4, 5]. Sleep duration is determined by how many hours an individual sleeps over 24 hours [6].

Recent studies have focused on exploring the methods of improving individual and environmental effects on disability related to inappropriate sleep duration by investigating its triggers and associations [7, 8]. The number of sleep hours has also been found to be associated with depression and socioeconomic factors [9]. For example, shorter sleep duration might lead to lower positive emotions and showed stronger associations with negative emotional affect [10, 11]. One question that needs to be asked, however, is the direction, nature, and strength of the association between these three variables respectively.

In the interest of studying sleep duration with the context of mental health status such as depression has been growing recently [12], the findings of some research papers suggest that self-reported sleep duration is a potential effect of depression [3, 13]. It was reported that people with depression have shorter sleep hours in contrast to people who were not depressed [14]. On the other hand, a bidirectional relationship between sleep duration and reported depression has been investigated [15]. Such studies are unsatisfactory because they do not explore this association with specific population demographics such as sociodemographic factors such as age, gender, and marital status. This paper focuses on sleep duration as effect of self-perceived depression after adjusting for sociodemographic factors.

Depression is defined as a set of disorders ranging from mild to moderate to severe [16]. It is widely recognized as a major public health problem worldwide [17]. Effect of depression has been extensively studied on the individual's daily functioning and productivity [18, 19]. To measure the level of depression in non-clinical populations; clinical and epidemiological studies have often used the established and validated eight-item Patient Health Questionnaire scale (PHQ-8) instead of the nine-item Patient Health Questionnaire scale (PHQ-9) [20]. As has been confirmed by previous studies, the scale can detect major depression with a specificity of 88%, to classify subjects into depressed or non-depressed, respectively [21–23]. Regionally, in Jordan, Lebanon, Syria and Afghanistan, the cutoff point of 10 was used [24–27]. Similarly, in the UAE, studies have mostly used the cutoff point of 10 [28–31].

In addition, there are some factors which are confounded with depression as predictors for insufficient sleep duration, such as age, gender, employment status, and marital status [32, 33]. For example, some research findings show that older adults are emotionally less affected by shorter sleep duration compared to younger ones [34]. Conversely, another study showed that older age was associated with shorter sleep duration after considering other factors such as presence of depression [35].

Moreover, gender differences in sleep duration were reflected on evidence and was linked to the situation of hormonal differences [36]. For example, some research findings reflected that females are more likely to perceive shorter sleep duration compared to males [37]. Other

studies showed significant association between marital status and sleep duration, in which single people reported shorter sleep duration compared to married people, in specific to those who reported perceiving and developing depressive symptoms [38, 39]. It was reported that worsening in depression symptoms among married individuals could lead to separation or being unmarried and lead to sleep disturbances [37, 40–42]. So, further exploration is required to study the interplay of these variables together.

In the UAE as a young country which underwent major living styles and lifestyle changes over the past 52 years, some chronic illnesses such as cardiovascular diseases, injury and depression were listed as the top causes of disability-adjusted life years in this country [43–46]. As a result, some researchers conducted the UAEHFS as a unique cohort study in the UAE and middle east region as it allows public health professionals and researchers to explore the relationship between common chronic diseases outcomes among Emirati adult population and related its risk factors or predictors [47].

In this cross-sectional study, we investigate the association between self-perceived sleep duration as an outcome and self-perceived depression (using the PHQ-8 as a screening measure for depression among Emirati adults) as a predictor, after adjustment for sociodemographic factors as age, gender, marital status, employment status, in a large UAEHFS cohort.

## Materials and methods

The United Arab Emirates Healthy Future Study (UAEHFS) is a population-based prospective cohort study recruited 15,000 adult Emiratis to explore the risk factors for the cardiovascular diseases in the UAE [47]. Opportunistic recruitment took place in different Emirates of the UAE and at multiple sites including health centers, universities and companies between February 2016 and March 2023 [47]. All participants were required to read and sign an informed consent. The exclusion criteria included those participants who reported any acute infection at the time of recruitment and pregnant women. This research was approved by the New York University in Abu Dhabi Research Ethics Committee (NYUAD/REC), (REF: 0072017R to [RA]) and the Institutional Review Board of Abu Dhabi Department of Health (DOH), (REF: DOH/ HQD/2020/516 to [RA]). A cross-sectional analysis of baseline cohort data from the 11,455 UAEHFS participants, was performed with Emirati nationals aged 18 years and above. The required data for this research purpose were accessed throughout the week of 19th June 2023.

### Measures

All Participants who signed the informed consent, answered a self-completed questionnaire, underwent physical measurements. The required demographic measurements such as age (18 years and above) and gender (male, female) were collected by the self-perceived questionnaire. Age was analyzed as continues variable.

Sleep duration was collected by asking the question "how many hours in total do you sleep including day and night?". Sleep duration was dichotomized into two sleep groups (sleep >7 and sleep <6), this is due to our expectation to have wide variation of sleep duration which starts from 3 hours per pay up to 21 hours per day since having variation in the individuals' sociodemographic factors [48, 49]. However, self-reported sleep duration was also analyzed as a continuous variable.

The Patient Health Questionnaire (PHQ-8) is a valid screening measure for depression [20, 23]. The following questions were included in the self-perceived questionnaire; for the last two weeks how often have you been bothered by any of the following problems? Little interest or pleasure in doing things, feeling down, depressed, irritable or hopeless, trouble falling or staying asleep, or sleeping too much, feeling tired or having little energy, poor appetite or

overeating, feeling bad about yourself–or that you are a failure or have let yourself or your family down, trouble concentrating on things, such as school work, reading or watching television and moving or speaking so slowly that other people could have noticed? or the opposite–being so fidgety or restless that you have been moving around a lot more than usual. The answers included not at all, several days, more than half the days, and nearly every day.

Since, the aim of this study is to investigate the association between self-perceived sleep duration and self-perceived depression but not the severity of depression, therefore, the total PHQ-8 score was analyzed as binary predictor to distinguish between depressed and non-depressed individuals using cutoff value of 10 (S1 Text) [20–22, 31]. A simpler binary classification of depression presence or absence can be more practical (S1 Text). This process refers to as us the "Binarization" of the PHQ-8 Score [20, 21].

Marital status groups categorized to (married, single), however, individuals with low frequencies, such as divorced, separated, widow and widower, were categorized as others [50]. Employment status was categorized as (employed, un-employed, and students), However, groups with low frequencies, such as retired, housewife, volunteering and unable to work because of sickness or disability, were categorized as others [51].

As an additional measure, we collected BMI by well-trained and experienced nurses, using Tanita MC-780 MA Segmental Body Composition Analyzer [52]. BMI was analyzed as continues variable.

Fig 1 shows the conceptual framework of self-reported sleep hours and its predictors (self-reported depression, sociodemographic factors).

## Statistical analysis

Summary statistics were presented by median and interquartile range (IQR) for continuous variables and frequency (%) for categorical variables. Chi-square test was used to compare proportions and Wilcoxon rank sum test was used to compare continuous predictors.

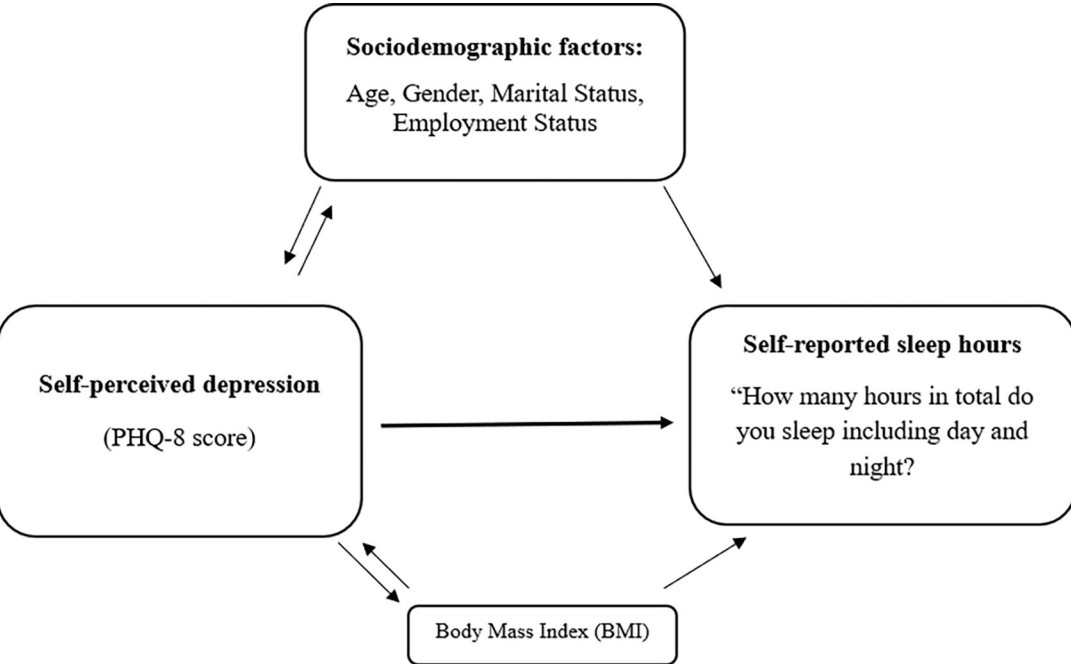

**Fig 1. Conceptual framework of self-reported sleep hours and its predictors (self-reported depression and sociodemographic factors).**

To examine the factors associated with self-reported sleep, univariate and multivariate logistic regression models were performed with the dichotomized self-reported sleep as the outcome. The predictors were Total PHQ-8 (continues), Age (continues), Gender (Females vs. Males), BMI (continues), and Marital status (categorical), employment status (categorical). Odds ratios (OR's) with 95% confidence intervals (CIs) were estimated with corresponding z-values and p-values [53].

In a secondary analysis, a multivariate linear regression model was conducted with sleep as a continues outcome. The predictors were Total PHQ-8 (continues), Age (continues), Gender (Females vs. Males), BMI (continues), and Marital status (categorical), employment status (categorical). The effect of each variable is estimated at 95% CI.

In sensitivity analysis, a multivariate logistic regression model was conducted using multiple imputations (see sensitivity analysis section). All applied statistical tests were two sided; p-value of less than 0.05 was considered statistically significant. Statistical analyses were performed in R version 4.3.1 [54].

## Sensitivity analysis

The primary statistical analysis included subjects with at least one non-missing value. However, in a sensitivity analysis, a multivariate imputation by chained equations (MICE) procedure was applied with classification and Regression Trees (CART) to impute missing values [55, 56]. 100 multiple imputations were used [57]. Rubin's rules were used to combine the multiple imputed estimates [58]. In each multiple imputation, a logistic regression model was performed with binarized sleep as the outcome. The predictors were Total PHQ-8 (linear), Age (linear), Gender (Females vs. Males), BMI (linear), and Marital status (categorical), employment status (categorical). Odds ratios (OR's) with 95% confidence intervals (CIs) were estimated.

The pattern of missing values was investigated, and it was found that subjects who "did not want to answer" were not systematically different from those who answered the questionnaire.

**Table 1. Number and percentages of the categorical variables by sleep categories.** Chi square p-values were computed for categorical variables and Wilcoxon rank sum test for continues variables. MS: Marital Status, ES: Employment Status.

| | Sleep ≤7 | | Sleep ≥6 | | Extreme (>4 & <13) | | | |
|---|---|---|---|---|---|---|---|---|
| | n | % | n | % | n | % | Total | p-value |
| Gender—Males | 1607 | 25.8 | 2680 | 43.1 | 1930 | 31.0 | 6,217 | 0.016 |
| Gender—Females | 958 | 25.6 | 1808 | 48.3 | 975 | 26.1 | 3,741 | |
| Gender—Missing | 2 | 0.1 | 8 | 0.5 | 1487 | 99.3 | 1,497 | |
| MS—Married | 1429 | 25.5 | 2833 | 50.6 | 1335 | 23.9 | 5,597 | <0.0001 |
| MS—Single | 1052 | 29.5 | 1490 | 41.7 | 1028 | 28.8 | 3,570 | |
| MS—Others | 68 | 25.5 | 122 | 45.7 | 77 | 28.8 | 2,67 | |
| MS—Missing | 18 | 0.9 | 51 | 2.5 | 1952 | 96.6 | 2,021 | |
| ES—Employed | 1412 | 32.3 | 2020 | 46.2 | 936 | 21.4 | 4,368 | <0.0001 |
| ES—Unemployed | 332 | 24.5 | 724 | 53.4 | 299 | 22.1 | 1,355 | |
| ES—Students | 435 | 25.5 | 1013 | 59.4 | 258 | 15.1 | 1,706 | |
| ES—Missing | 388 | 9.6 | 739 | 18.4 | 2899 | 72.0 | 4,026 | |
| | Median (IQR) | | Median (IQR) | | Median (IQR) | | | |
| Total PHQ-8 | 5 (2, 9) | | 4 (1, 8) | | 5 (1, 9) | | | <0.0001 |
| Age | 27 (22, 34) | | 25 (21, 32) | | 27 (22, 35) | | | <0.0001 |
| BMI | 26.9 (23.2, 30.5) | | 25.8 (22.3, 30.1) | | 26.5 (23.1, 30.5) | | | <0.0001 |

Therefore, "prefer not to answer" was recorded as missing value in the statistical analysis and was considered a missing variable in the sensitivity analysis [55].

## Results

11,455 participants were included in the statistical analysis (Table 1). Of 11,455 participants, 4,295 (37.5%) were included in the complete case analysis after omitting missing values (Table 2). Males were older than females in this cohort with median of 28 (IQR:23–36), and 24 (IQR: 20–34). Table 1 illustrates the frequencies and percentages of categorical variables within sleep categories. Both males and females reported higher percentage of sleep >6 hours, 43.1% and 48.3%, chi square p-value = 0.016 when investigated with reported depression as a predictor. Similar patterns were observed by the marital status and employment status categories. Married people reported higher percentage of sleep hours duration of 50.6% as compared to single (41.7%) and others (45.7%), chi square p-value <0.0001, when investigated with reported depression as a predictor. Students and unemployed participants have reported higher percentage of longer sleep hours 59.4% and 53.4% as compared to employed 46.2%, chi square p-value <0.0001 (Table 1), when investigated with reported depression as a predictor.

Table 2 shows the estimated odds ratios (95% CIs), z-value, p-value from fitted univariate and multivariate logistic regression model associated with self-reported Sleep. The total PHQ-8 was statistically significant negatively associated with sleep categories OR of 0.965 (95% CI: 0.954, 0.976) and 0.961 (95% CI: 0.948, 0.974) in univariate and multivariate logistic regression model respectively. Similarly, age and BMI were also statistically significant negatively associated with sleep categories in participants data, when associated with reported depression, OR of 0.975 (95% CI: 0.969, 0.981), 0.979 (95% CI: 0.972, 0.987) in the univariate logistic regression model correspondingly. The result remains statistically significant in the multivariate logistic regression model, OR of 0.979 (95% CI: 0.969, 0.990), 0.987 (95% CI: 0.977, 0.998) individually. Furthermore, the employment status was also statistically significant in the univariate and multivariate logistic regression model, OR of 1.524 (95% CI: 1.316, 1.765) and 1.628 (95% CI: 1.427, 1.856) for unemployed and students in univariate model respectively (Table 2). Similarly, OR was 1.367 (95% CI: 1.138, 1.643) and 1.238 (95% CI: 1.021, 1.501) for unemployed and students in multivariate model respectively (Table 2). The gender was only statistically significant in the univariate logistic regression model with female having higher sleep hours, OR of 1.132 (95% CI: 1.024, 1.250). No statistically significant association was observed between sleep categories, reported depression, and marital status and in the univariate and multivariate logistic regression model (Table 2).

**Table 2. Estimated odds ratios (95% CIs), z-value, p-value from fitted univariate and multivariate logistic regression model associated with self-reported sleep.** MS: Marital Status, ES: Employment Status.

| | Univariate | | | | | Multivariate (n = 4295, 2718 sleep>6) | | |
|---|---|---|---|---|---|---|---|---|
| | OR (95% CI) | z-value | p-value | N | Sleep >6 | OR (95% CI) | z-value | p-value |
| Total PHQ-8 | 0.965 (0.954, 0.976) | -6.351 | <0.0001 | 5,432 | 3,477 | 0.961 (0.948, 0.974) | -5.951 | <0.0001 |
| Age | 0.975 (0.969, 0.981) | -7.862 | <0.0001 | 6,533 | 4,132 | 0.979 (0.969, 0.990) | -3.837 | <0.0001 |
| Gender—Female | 1.132 (1.024, 1.250) | 2.429 | 0.015 | 7,053 | 4,488 | 0.987 (0.854, 1.141) | -0.176 | 0.860 |
| BMI | 0.979 (0.972, 0.987) | -5.094 | <0.0001 | 6,533 | 4,132 | 0.987 (0.977, 0.998) | -2.377 | 0.017 |
| ES—Employed | Reference group | | | 5,936 | 3,757 | Reference group | | |
| ES—Unemployed | 1.524 (1.316, 1.765) | 5.635 | <0.0001 | | | 1.367 (1.138, 1.643) | 3.339 | 0.001 |
| ES—Students | 1.628 (1.427, 1.856) | 7.272 | <0.0001 | | | 1.238 (1.021, 1.501) | 2.176 | 0.030 |
| MS—Married | Reference group | | | 6,994 | 4,445 | Reference group | | |
| MS—Others | 0.714 (0.646, 0.791) | -6.502 | <0.0001 | | | 0.980 (0.824, 1.165) | -0.228 | 0.820 |
| MS—Single | 0.905 (0.668, 1.226) | -0.645 | 0.519 | | | 1.486 (0.993, 2.223) | 1.927 | 0.054 |

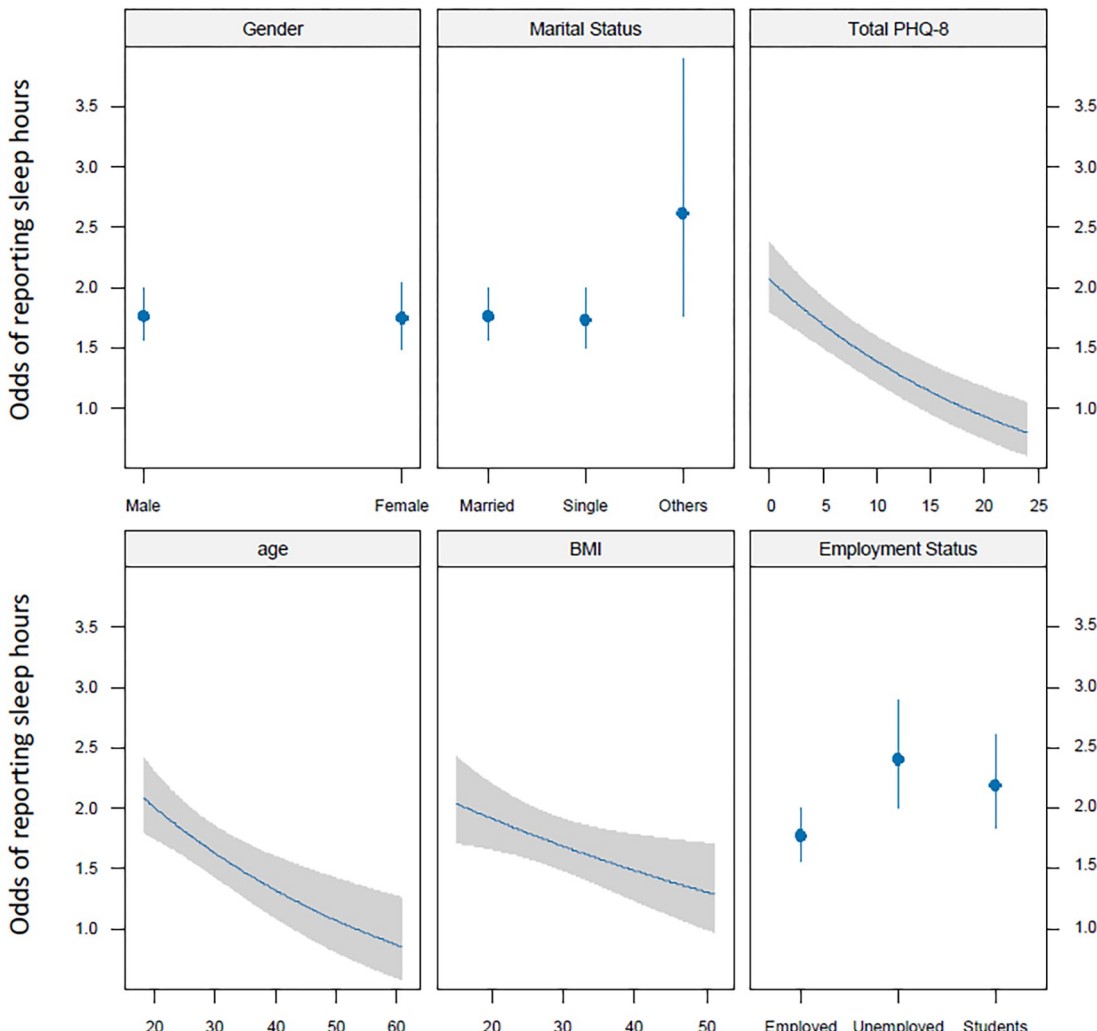

**Fig 2. Predicted odds and confidence bands of self-reported sleep from the fitted multivariate regression model in the primary analysis, refer to (Table 2).**

Fig 2 shows the predicted odds and confidence bands of self-reported sleep for six predictors included in the fitted multivariate regression model in the primary analysis in Table 2. Increasing the values of the total PHQ-8, age and BMI, leads to decrease in the odds of normal self-reported sleep duration hours (Fig 2). Marital status with category (others) had higher odds of reporting shorter sleep duration as compared to single and married in association with depression. Employed participants had lower odds of reporting shorter sleep duration as compared to unemployed and students' participants (Table 1), in association with depression as one of the predictors. However, there was no statistically significant difference between males and females in terms of reported sleep duration.

Fig 3 shows summary of the effect predictors in a fitted multivariable linear regression model with self-reported sleep as continues outcome, from the secondary analysis. Total PHQ-8, Age, BMI was statistically significant. Students and unemployed have a higher effect of reporting higher sleep hours compared to employed.

S1 Table shows the estimated odds ratios (95% CIs), z-value, p-value from 100 multiple imputations using Rubin's rules. Similar results are observed as in the primary analysis, where

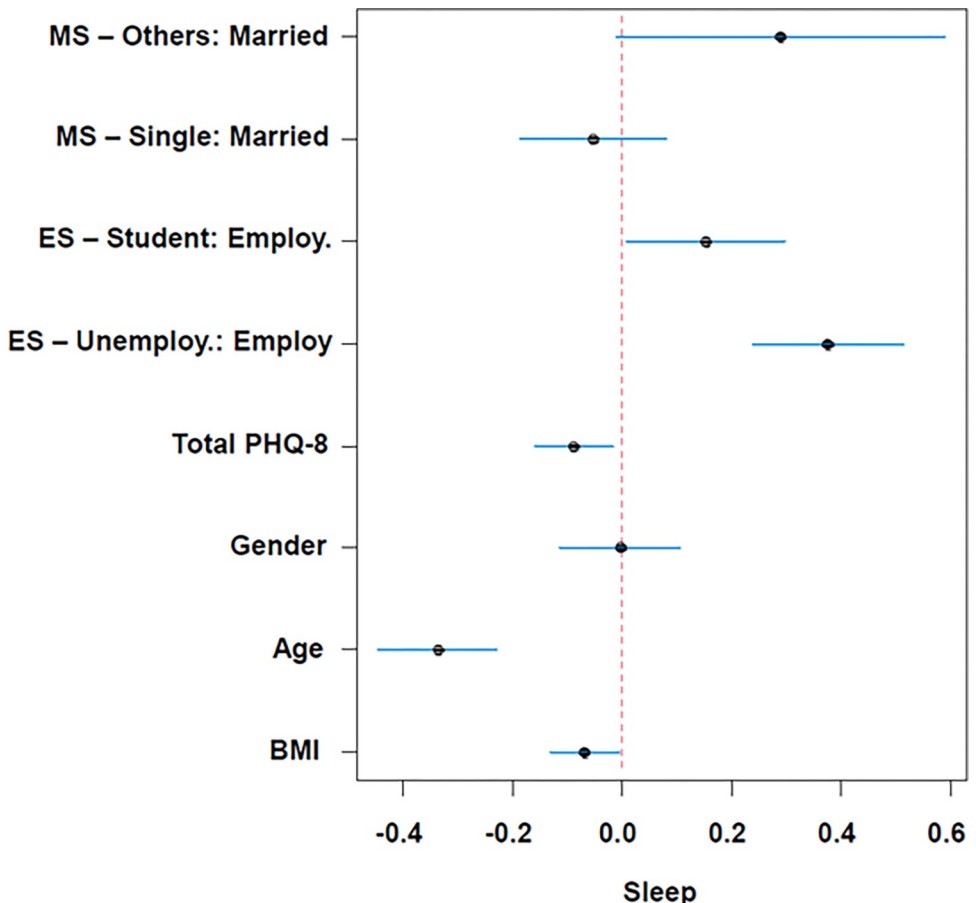

**Fig 3. Summary of the effect predictors in a fitted linear regression model with a sleep as continues outcome.** MS: Marital Status, ES: Employment Status.

total PHQ-8, age and BMI were statistically significantly negative associated with high self-reported sleep hours, OR of 0.966 (95% CI: 0.957, 0.976), 0.981 (95% CI: 0.974, 0.989), and 0.989 (95% CI: 0.981, 0.996), respectively. Similarly, there was no statistically significant difference between males and females. Unemployed and students were more likely to report higher sleep hours as compared to employed, OR 1.334 (95% CI: 1.161, 1.533), and 1.262 (95% CI: 1.083, 1.47), respectively. There was no statistically significant difference between the categories of marital status in reporting sleep hours (S1 Table).

## Discussion

Findings of this study revealed that individuals with higher PHQ-8 scores have a higher risk of reporting shorter sleep duration. However, participants with lower PHQ-8 scores reported normal sleep hours duration. Previous studies' findings showed similar outcomes between longer sleep duration and chronic diseases such as depression [59, 60]. Additionally, evidence from literature showed a significant improvement in sleep duration in individuals who undergo treatment for depression. This supports the significant association between self-perceived sleep duration and self-perceived depression which was reflected in our study.

In addition, this study examined the association between self-perceived sleep duration and depression after adjustment for age, gender, and sociodemographic factors. The results showed

that increasing age may decrease the risk of reporting short sleep duration in association with reported depression. Another study found a significant relationship between sleep duration, depression, and age in which older adults reported less odds for depression and more odds for normal sleep duration [60]. This could be explained with the ability of older adults to control feelings and use wisdom in daily life [61]. In general, the age effect in depressed people is evidenced to be significant with sleep duration and this supports our findings [60, 61].

Moreover, this study investigated the effect of gender on the nature of the association between self-perceived sleep duration and reported depression. The findings reflect that, in association with reported depression, both males and females exhibit no difference in reporting sleep hours. Available evidence supports this finding [62]. Other evidence reflected only one hour difference in sleep duration between both genders when the depression is the predictor [63]. This supports the findings of our study in which no difference in gender in terms of self-perceived sleep duration when associated with depression. However, another study has evidenced that females are more likely to report depression compared to males, but it didn't investigate the sleep duration as an outcome [62]. Therefore, we recommend future research in order to have a better understanding of the nature of the effect of gender on the association between self-perceived sleep duration and reported depression.

Also, there were statistically significant differences observed between different marital status and different employment status categories among analyzed study participants data when examined their effect on the association between self-perceived sleep duration and depression. Married people reported a higher percentage of sleep hours duration compared to single and other participants when associated with reported depression. Also, students and unemployed participants have reported longer sleep hours as compared to employed ones when associated with reported depression as one of the predictors. However, these findings are similar to what was observed before in similar research [64, 65]. Further research is recommended to reflect better understating of these associations in order to be able to apply it on the public health promotion efforts.

As an additional measure, Body Mass Index (BMI) was included in the statistical analysis. It has been evidenced to associate with sleep duration and plays a role in the relationship between sleep duration and depression [64, 66]. The negative association between BMI and self-perceived sleep duration is well-documented in sleep and obesity research [67, 68]. This association suggests that as BMI increases, self-perceived sleep duration tends to decrease in the depressed individuals [66]. We had similar findings in our study. In depressed participants, who had increased BMI, they reported shorter sleep duration.

Regarding the categorization of sleep, we found evidence in literature that supports the decision we made to put it in two main categories of >6 hours and <7 hours sleep duration [69, 70].

The method of dividing sleep duration into binary groups—those who report less than 7 hours and those who report 7 hours or more—is a widely accepted approach in sleep research. Recommendations from numerous health organizations are incorporated into this binary classification. For example, the National Sleep Foundation advocates 7 to 9 hours of sleep for adults to maintain optimal health. This suggests that at least 7 hours is generally considered a crucial threshold for adequate sleep, supporting its use as a cutoff point in sleep studies [48, 71]. However, a multivariate linear regression model was also used to analyze sleep duration as continuous variable (Fig 2).

Also, the participants who reported extreme sleep durations, either less than 4 or more than 13 hours per day, were classified into an 'Extreme' category due to the unusual sleep patterns [72]. Additionally, responses indicating "I don't know" or "Prefer not to answer," were also categorized as 'Missing'. This allows us to maintain clear and meaningful categories for our analysis while accounting for all responses. For other variables, missing values were categorized as missing categories and included in the multivariate regression analysis [73].

### Strength and limitations

Although there were variations on the significance of the association between self-perceived sleep duration and self-perceived depression after considering sociodemographic factors, the results of this study are unique insights specific to this geographic region as part of the world as it is one of the first and biggest cohort national studies in the Gulf area. The findings of this study have a potential value for researchers and public health professionals as it presents novel data on the PHQ-8 score in a healthy UAE population, which has not been explored before. Furthermore, the results of this study can help contribute to the knowledge base on current and potential population mental health impact in the UAE and Gulf Region. The major key message derived from this research is that prevention and treatment of depression may contribute to improve the duration of sleep hours. Moreover, it is very important to consider sociodemographic factors effects in the association between sleep duration and depression and apply that within the public health promotion policies and activities in the country.

One major limitation of this study was the big number of missing values with some variables. This could be improved in future research by working on measures to improve the participant responses to the self-perceived questionnaire. Despite certain limitations of this study, our multiple imputation makes our analysis more reliable, stronger, and flexible. The data collected on the PHQ-8 score of a healthy UAE population is groundbreaking and of significant value to researchers and healthcare professionals alike, as this area has remained unexplored.

### Conclusion

The study findings indicate a correlation between depression and an increased probability of individuals reporting shorter self-perceived sleep durations especially when considering the sociodemographic measures factors. There was a variation in the effect of depression on sleep duration among different study groups. Married people reported longer sleep duration as compared to single and unmarried ones when examined reported depression as a predictor. Finally, students and unemployed individuals have reported longer sleep hours as compared to employed participants when associated with reported depression. There was no difference in gender self-perceived sleep duration when associated with reported depression. The age effect is evidenced to be significant on sleep duration when associated with reported depression. Further research is recommended to better understand the variations of the effect of sociodemographic factors and reported depression on reported sleep duration.

### Supporting information

**S1 Table. Estimated odds ratios (95% CIs), z-value, p-value from 100 multiple imputations using Rubin's rules.** MS: Marital Status, ES: Employment Status.
(DOCX)

**S1 Text. Binarization of the PHQ-8 score.**
(DOCX)

### Acknowledgments

The authors would like to thank all members of the Public Health Research Center (PHRC) at New York University Abu Dhabi for active involvement in the United Arab Emirates Healthy Future study.

## Author Contributions

**Conceptualization:** Abdishakur Abdulle, Raghib Ali.

**Data curation:** Mitha Al Balushi, Sara Al Balushi.

**Formal analysis:** Mitha Al Balushi, Amar Ahmad, Sara Al Balushi.

**Methodology:** Mitha Al Balushi, Amar Ahmad, Abdishakur Abdulle, Raghib Ali.

**Project administration:** Amar Ahmad, Abdishakur Abdulle, Raghib Ali.

**Supervision:** Amar Ahmad, Sayed Javaid, Fatma Al-Maskari, Abdishakur Abdulle, Raghib Ali.

**Visualization:** Abdishakur Abdulle, Raghib Ali.

**Writing – original draft:** Mitha Al Balushi, Amar Ahmad, Sara Al Balushi.

**Writing – review & editing:** Mitha Al Balushi, Abdishakur Abdulle.

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
