## [Editor Report · Decision Letter 0]

10 Sep 2023

PGPH-D-23-01523

Investigating the association between self-reported sleep duration and self-reported depression among United Arab Emirates Healthy Future Study participants

Dear Dr. Albalushi,

Thank you for submitting your manuscript to PLOS Global Public Health. After careful consideration, we feel that it has merit but does not fully meet PLOS Global Public Health’s publication criteria as it currently stands. Therefore, we invite you to submit a revised version of the manuscript that addresses the points raised during the review process.

We look forward to receiving your revised manuscript.

Kind regards,

Thang Van Vo, PhD

Academic Editor

Journal Requirements:

2. Please provide separate figure files in .tif or .eps format only and remove any figures embedded in your manuscript file. Please also ensure all files are under our size limit of 10MB.

Additional Editor Comments (if provided):

This manuscript is meeting in basic requirement of the Plos Global Health Journal, however, before it will be proceeding for having peer review the authors should have some revision upon our comments as follows:

Structure of the manuscript:

To be consistent with the abstract, authors is required to add conclusion section in the full-text manuscript

The title:

It should be revised how to cover and be consistent with the aim of study that not only investigating the association between self-reported sleep duration and self-reported depression but also measuring possible association between sleep duration and other sociodemographic factors and BMI... Be sure that sleep duration was dependent variable as outcome in cause- effect relationship o a cross sectional study in the first baseline survey.

Introduction section:

There are still controversial to assume that which one among depression or sleep disorder is dependent or independent variable in cause-effect relationship assumption? Therefore, authors may clarify better in this study that sleep duration of study is considered as outcome variable and probably affected by depression and the others by adding more literature review

Measures section:

The way of classifying variable of sleep duration as category group seems not convincing enough for data analysis. It's much better to classify sleep duration into binary variable having 2 groups (normal/disorder) then the findings would be stronger in statistical analysis.
---

## [Decision Letter · Decision Letter 1]

31 Jan 2024

PGPH-D-23-01523R1The association between self-reported sleep duration and depression adjusted for sociodemographic factors: A cross sectional study from the United Arab Emirates.PLOS Global Public Health

Dear Dr. Albalushi,

Thank you for submitting your manuscript to PLOS Global Public Health. After careful consideration, we have decided that your manuscript does not meet our criteria for publication and must therefore be rejected.

We are sorry that we cannot be more positive on this occasion. We very much appreciate your wish to present your work in one of PLOS's Open Access publications. Thank you for your support, and we hope that you will consider PLOS Global Public Health for other submissions in the future.

Yours sincerely, Marianne Clemence,Staff Editor, on behalf of, Thang Van Vo, PhD

Academic Editor

Additional Editor Comments (if provided):

Academic Editor's decision to reject this Manuscript for publication

Reason:

The content of this manuscript does not meet the standards of rigor required by the journal to be considered for publication.

Comment:

We are very sorry to say that this submission does not meet the standards required for publication in PLOS Global Health Journal and that because of the major limitation of manuscript which would also take considerable time to address. Your research did not achieve the objectives and methods of data analysis were not justified to provide strong evidences to share with academic readers over the world as this Journal expected. In addition, your analysis, discussions and also conclusion did not strongly support the objectives as formulated.

Two reviewers have rejected from the review process because of fundamental errors and what they see as the inability of the authors to revise the manuscript appropriately. Thus is appears that at present, it is best to reject the manuscript in its current form as it does not meet the standards of rigor of the journal.

We wish you the very best in undertaking an extensive revision of the manuscript with a view to a re-submission or a submission elsewhere.

Reviewers' comments:

Reviewer's Responses to Questions

**Comments to the Author**

1. If the authors have adequately addressed your comments raised in a previous round of review and you feel that this manuscript is now acceptable for publication, you may indicate that here to bypass the “Comments to the Author” section, enter your conflict of interest statement in the “Confidential to Editor” section, and submit your "Accept" recommendation.

Reviewer #1: (No Response)

Reviewer #2: (No Response)

Reviewer #3: (No Response)

2. Does this manuscript meet PLOS Global Public Health’s publication criteria? Is the manuscript technically sound, and do the data support the conclusions? The manuscript must describe methodologically and ethically rigorous research with conclusions that are appropriately drawn based on the data presented.

Reviewer #1: Partly

Reviewer #2: (No Response)

Reviewer #3: Partly

3. Has the statistical analysis been performed appropriately and rigorously?

Reviewer #1: No

Reviewer #2: (No Response)

Reviewer #3: Yes

4. Have the authors made all data underlying the findings in their manuscript fully available (please refer to the Data Availability Statement at the start of the manuscript PDF file)?

Reviewer #1: Yes

Reviewer #2: (No Response)

Reviewer #3: No

5. Is the manuscript presented in an intelligible fashion and written in standard English?

Reviewer #1: No

Reviewer #2: (No Response)

Reviewer #3: Yes

6. Review Comments to the Author

Reviewer #1: The aim of the study is to “investigate the relationship between self-reported sleep hours and the eight items patient health questionnaire (PHQ-8) as a screening measure for depression among Emirati adults”. However, I am not sure the authors achieved this aim. In particular, I feel like the statistical methods were overly complicated and poorly justified. Given that previous research has found correlations between depression and both too little and too much sleep, I don’t understand why the authors did not choose to treat sleep duration as a continuous measure? I would have like to have seen a regression or at least a scatter plot of PHQ-8 score and sleep hours, as I imagine the relationship may have been u shaped (this is indicated in the box blots the authors included). If there was a good reason for grouping the sleep categories in the way they did, I would have needed more justification for this to make sense. In any case, I find multinomial regression extremely complicated to explain and I generally try to avoid it. Given this data and the authors’ aims, I would have thought some sort of linear or non-parametric regression would be easier to explain. If the hypothesis is that the relationship between depression and sleep is moderated by age/gender/BMI, stratified analysis or the inclusion of an interaction term would have made more sense. I read the paper several times, but I still don’t entirely understand the analysis. More minor points, I think using scientific notation for p values is confusing and prefer to see “p<.0001” in these situations. Finally, there are some typos in the text, for example in the abstract the numbers of males and females are not included, it says “n males and n females” (page 4 line 27).

Reviewer #2: • In the abstract, the reports of conclusions do not seem aligned with the objective and the title. The aim in the abstract/introduction is to investigate the relationship between self-reported sleep hours and self-perceived depression adjusting for sociodemographic factors, but analyses, discussion, and conclusions seem disconnected from this objective.

• In the abstract, there are also several findings summarised in the conclusions that were not even reported in the results section. Additionally, I believe it would be beneficial to include descriptive data of the participants included in the final analysis (N=4,295) rather than descriptive data out of 11,455 participants.

• The introduction suggests sleep duration as a risk factor of depression, but it is unclear why this paper “focuses on sleep duration as effect of self-perceived depression (lines 62-63)”. Also, due to the cross-sectional study design, the paper can only analyse the association between sleep duration and depression- causal inference cannot be determined.

• Introduction/description of confounders (e.g., age, sex, BMI) in the association between depression and sleep is confusing. For these factors to be confounders, authors need to explain clearly based on the current evidence that they are associated with both the outcome and the exposure, but not on the causal pathway. The use of ‘interactive relationship’ and ‘interaction’ in the introduction is also confusing.

• Discussion should be written better and be compared with the existing literature.

• Measures: more information on the outcome/exposure should be provided, e.g., how they were measured and what specific questions were used.

• Authors should discuss limitations, e.g., cross-sectional study design, sampling strategy, bias in self-reported outcome and exposure.

• Analysis section: labelling variable as ‘linear’ is strange- replace by ‘continuous’.

• Another concern I have is regarding the treatment of missing values as unknown categories in the regression analysis (lines 247-248). It seems unusual to include them in the analysis.

Reviewer #3: Title: The title may need modification to let readers know first-hand what the study entails. Consider a title like ‘’Sociodemographic predictors of the association between self-reported sleep duration and depression.”

Abstract (Page 1): The conclusion part of the abstract should reflect the titles. Consideration may be given only to those socio-demographic factors that correlated positively.

2nd Abstract (Page 4): Under “Result”, there were no values for percentages (%) for males and females. Incomplete statement noted within the third statement.

Introduction: Line 56; there should be a clear definition of depression. What type of disorder is depression?

Materials and Method:

Under “Measures”, age was categorized as follows; 18 – 19, 20 -29, 30 – 39, and 40+. What is the reason for using 40+ instead of 40 – 49, 50 – 59 etc.

Under “Result””

Table 1: The table is easy to understand by scientists.

Figure 1A, 1B, and 1C: What are the reasons for using boxplot?

Is there any table or figure that shows the relationship between sleep hours and depression?

Discussion and Conclusion:

The conclusion does not highlight and also reflects the title of the paper.

7. PLOS authors have the option to publish the peer review history of their article (what does this mean?). If published, this will include your full peer review and any attached files.

**Do you want your identity to be public for this peer review?** For information about this choice, including consent withdrawal, please see our Privacy Policy.

Reviewer #1: No

Reviewer #2: No

Reviewer #3: No

---

## [Editor Report · Decision Letter 2]

16 Apr 2024

PGPH-D-23-01523R2

Sociodemographic predictors of the association between self-reported sleep duration and depression

Dear Dr. Albalushi,

Thank you for submitting your manuscript to PLOS Global Public Health. After careful consideration, we feel that it has merit but does not fully meet PLOS Global Public Health’s publication criteria as it currently stands. Therefore, we invite you to submit a revised version of the manuscript that addresses the points raised during the review process.

We look forward to receiving your revised manuscript.

Kind regards,

Thang Van Vo, PhD

Academic Editor

Journal Requirements:

Additional Editor Comments (if provided):

Thanks for your patient to take a lot of time for very serious revision of manuscript. We re-consider your latest manuscript was revised carefully upon reviewers' comments on research methodology (clearer variable clarification and statistical data analysis...). It's very much appreciated with a stronger evidence of the findings of study on "Sociodemographic predictors of the association between self-reported sleep duration and depression". Before we will make a final decision for possible publication, the author should add a conceptual framework of your study, in your Measures section of the full text, to clarify better the cause-effect relationship between study variables (The predictors of "self-reported sleep hours" variable are the self-reported depression and other factors as described).
---

## [Editor Report · Decision Letter 3]

30 Apr 2024

Sociodemographic predictors of the association between self-reported sleep duration and depression

PGPH-D-23-01523R3

Dear Dr Albalushi,

We are pleased to inform you that your manuscript 'Sociodemographic predictors of the association between self-reported sleep duration and depression' has been provisionally accepted for publication in PLOS Global Public Health.

Best regards,

Thang Van Vo, PhD

Academic Editor